# Axial Vibration Characteristics of Fluid-Structure Interaction of an Aircraft Hydraulic Pipe Based on Modified Friction Coupling Model

**Lingxiao Quan [1,2], Shichao Che [1], Changhong Guo [1,*], Haihai Gao [1] and Meng Guo [3]**

[1] School of Mechanical Engineering, Yanshan University, Qinhuangdao 066004, Hebei, China; lingxiao@ysu.edu.cn (L.Q.); shichaoche@gmail.com (S.C.); gaohaihai.ysu.edu.cn@stumail.ysu.edu.cn (H.G.)

[2] Hebei Provincial Key Laboratory of Heavy Machinery Fluid Power Transmission and Control, Yanshan University, Qinhuangdao 066004, Hebei, China

[3] School of Mechanical Engineering, Beijing Institute of Technology, Beijing 100081, China; guomeng@bit.edu.cn

[*] Correspondence: guochanghong@ysu.edu.cn

**Abstract:** This paper aims at studying the axial vibration response characteristics of Fluid-Structure Interaction (FSI) vibration of aircraft hydraulic pipe when considering the friction coupling. Based on the Brunone empirical model and the Zielke weighting function, an expression of fluid shear stress of the hydraulic pipeline is presented for a wide range of Reynolds number, and the friction model of the FSI 14-equation for high-speed and high-pressure hydraulic pipeline is modified. On this basis, a left-wing hydraulic pipeline of the C919 airplane is taken as the verification object and modeled, then the FSI vibration 14-equation is solved using frequency-domain transfer matrix method in MATLAB to analyze the modal and the axial vibration characteristics of the pipeline. Ultimately, a test experiment is given and discussed by being compared with the numerical simulation results, which confirmed the correctness of the friction model and demonstrated that the analytic accuracy of axial velocity response of FSI vibration could be improved by considering the friction coupling.

**Keywords:** hydraulic pipeline; friction coupling; axial vibration; fluid-structure interaction

## 1. Introduction

High-speed, high-pressure and high power-to-weight ratio are the strategic development direction of modern aircraft hydraulic system, which can reduce the weight and volume of aircraft hydraulic system effectively, but the Fluid-Structure Interaction (FSI) vibration of aircraft hydraulic pipeline will be enhanced at the same time. Moreover, the aircraft hydraulic pipeline system is subjected to superposition effect and coupling action of pressure, temperature, body deformation, vibration, and acceleration loading, which causes the flow state of the fluid in the pipeline to be extremely complicated. For instance, over 21% of about 800 hydraulic pipelines have a higher Reynolds number than the critical Reynolds number of 2320 in all hydraulic pipelines of Advanced Regional Jet for the 21st Century (ARJ21-700) aircraft developed by Commercial Aircraft Corporation of China Ltd. (COMAC), which shows that turbulence phenomenon exists in the pipelines [1]. When the fluid is in a laminar flow state, the coupling vibration between the fluid and the pipe has been studied extensively. However, the turbulence is a random, unsteady three-dimensional flow with strong nonlinearities, making it hard to obtain a friction model between the fluid and the tube wall. Hence, establishing a more accurate friction model to study FSI vibration characteristics of high-speed high-pressure hydraulic pipelines is of significant theoretical value and practical significance.

Numerous researchers have proposed and improved FSI vibration models to investigate the vibration characteristics of hydraulic pipeline. Joukowsky presented water hammer theory, which laid a theoretical foundation for the research of the Fluid-Structure Interaction mechanism of pipeline [2]. Skalak expanded water hammer theory and given the 4-equation model [3]. Tijsseling and Davidson perfected the 4-equation model. Besides, Davidson proposed the 8-equation model and solved it using frequency-domain transfer matrix method [4,5]. Walker presented the 6-equation model without considering poisson coupling [6]. Wiggert put forward the 14-equation model ignored friction coupling and bourdon coupling [7]. On the basis of Wiggert's 14-equation model, Tentarelli consummated the 14-equation model and verified the effectiveness of this method by testing [8]. Keramat and Kutin further improved the FSI vibration model, which makes the calculated value more accurate [9,10]. In addition, various numerical and analytical methods were developed to predict FSI vibration. Frequency-domain transfer matrix method of 14 equation model was discussed and verified in detail by Zhang Lixiang [11,12]. Liu Gongmin analyzed the FSI vibration of the pipeline under different support conditions and proposed the 14-equation model based on the elastic support condition [13]. A frequency modeling and solving method considering complex constraints is proposed to analyze the vibrational characteristics of complex pipelines by Xu Yuanzhi [14].

Numerous scholars have investigated friction coupling models. The first group of models was given by Zielke in 1968, which is applicable for transient laminar flow and has a great calculation precision [15]. In this model, the friction term will be influenced by instantaneous flow acceleration or the past velocity changes. Karam T.J. obtained a simplified friction model by adding a weight function into the Zielke model, but calculation accuracy of this model is decreased [16]. Vardy and Brown postulated that the fluid inside the pipeline split into laminar flow layer and turbulent flow layer and Zarzycki assumed that the fluid inside the pipeline separated into four layers, in order to develop a model that could be applied to a wide range of Reynolds numbers and frequencies [17–20]. Apart from the models above, another group of empirical formula model was proposed by Daily, he thought the shear friction force depended on the velocity of unsteady flow and local instantaneous acceleration [21]. This model was verified and improved by the subsequent researchers, for instance, through calculations and deducing, Carstens derived the expression of turbulent unsteady shear stress [22]. Brunone combined the instantaneous local acceleration and convective acceleration with friction coupling, obtained the classical Brunone model [23,24]. Vitkovsky studied the application of Brunone model under different flow conditions, and the convection acceleration of the fluid is considered modifying the model [25]. However, an obvious shortcoming for the above models is that the empirical coefficient can be hardly determined.

In the present study, the FSI 14-equation model for an aviation hydraulic pipe is built and the Zielke model and Brunone model are used to modify the fluid shear friction stress model of the axial dynamic model. Hence, a new shear friction term that applicable to a wide range of Reynolds numbers and frequencies is formed. The frequency-domain transfer matrix method is applied to resolve the FSI 14-equation model to analyze the modal and axial vibration velocity of the above model. Then, two experiments are given to verify the correctness of modal analysis and to compare the axial vibration velocity when considering the friction term or ignore the friction term. The results show the model is correct and indicate that consider the friction term is helpful to improve the analytic accuracy of axial vibration velocity.

## 2. Theoretical Model

### 2.1. Fluid-Structure Interaction (FSI) 14-Equation Model of Hydraulic Pipeline

The FSI dynamic 14-equation model of hydraulic pipeline is mainly based on the water hammer theory, using classical beam equations theory to describe the kinetic relationship between pipeline and fluid. The model is built based on the following basic assumptions: (1) the velocity of the sound wave is greater than the flow velocity; (2) the wavelength of the sound wave is longer than the pipeline

diameter; (3) only elastic deformation is occurred in the pipeline wall; (4) the hydraulic oil is a kind of Newtonian fluid; (5) no separation and cavitation in the fluid during the movement process. Figure 1 shows the FSI dynamic 14-equation model of the bent pipeline microelement, which consists of two hydrodynamic equations and twelve pipeline equations [26].

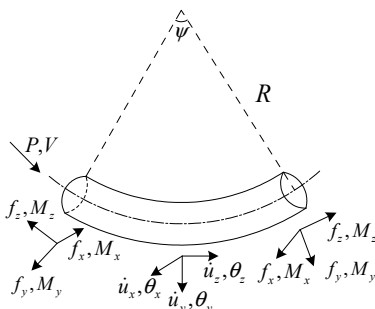

**Figure 1.** The schematic diagram of the bent pipeline element.

(1) Axial dynamic model

The axial dynamic model has two components: the water hammer equation of fluid and the axial dynamic equilibrium equation of pipeline, and it describes the transient behavior of the fluid and the pipeline.

$$\frac{\partial P}{\partial z} = -\rho_f \frac{\partial V}{\partial t} - \frac{2\tau_0}{R\rho_f} \tag{1}$$

$$\frac{\partial V}{\partial z} = -\frac{1}{K^*}\frac{\partial P}{\partial t} - 2v\frac{\partial \dot{u}_z}{\partial z} - \frac{\dot{u}_y}{R} \tag{2}$$

$$\frac{\partial f_z}{\partial z} = -A_p\rho_p\frac{\partial \dot{u}_z}{\partial t} - \frac{f_y}{R} \tag{3}$$

$$\frac{\partial \dot{u}_z}{\partial z} = \frac{\partial f_z}{EA_p\partial t} - \frac{vr}{Ee}\frac{\partial P}{\partial t} - \frac{\dot{u}_y}{R} \tag{4}$$

(2) Transverse y-z dynamics model

This model describes the vibration of the bent pipeline microelement by the force and moment balance equation in the y-z transverse direction.

$$\frac{\partial f_y}{\partial z} = -\left(\rho_f A_f + \rho_p A_p\right)\frac{\partial \dot{u}_y}{\partial t} + \frac{f_z}{R} + \frac{A_f}{R}P \tag{5}$$

$$\frac{\partial \dot{u}_y}{\partial z} = \frac{1}{k^2 GA_p}\frac{\partial f_y}{\partial t} - \dot{\theta}_x + \frac{\dot{u}_z}{R}, \ k^2 = 2\frac{1+v}{4+3v} \tag{6}$$

$$\frac{\partial M_x}{\partial z} = -\left(\rho_p I_p + \rho_f I_f\right)\frac{\partial \dot{\theta}_x}{\partial t} + f_y \tag{7}$$

$$\frac{\partial \dot{\theta}_x}{\partial z} = \frac{1}{EI_p/ff}\frac{\partial M_x}{\partial t} \tag{8}$$

(3) Transverse x-z dynamics model

This model describes the vibration of the bent pipeline microelement by the force and moment balance equations in the x-z transverse direction.

$$\frac{\partial f_x}{\partial z} = -\left(\rho_f A_f + \rho_p A_p\right)\frac{\partial \dot{u}_x}{\partial t} \tag{9}$$

$$\frac{\partial \dot{u}_x}{\partial z} = -\frac{1}{k^2 GA_p}\frac{\partial f_x}{\partial t} + \dot{\theta}_y, \ k^2 = 2\frac{1+v}{4+3v} \tag{10}$$

$$\frac{\partial M_y}{\partial z} = -\left(\rho_p I_p + \rho_f I_f\right)\frac{\partial \dot{\theta}_y}{\partial t} - f_x + \frac{M_z}{R} \tag{11}$$

$$\frac{\partial \dot{\theta}_y}{\partial z} = \frac{1}{EI_p/ff}\frac{\partial M_y}{\partial t} + \frac{\dot{\theta}_z}{R} \tag{12}$$

(4) Torsion dynamics model

This model describes the torsional vibration of the bent pipeline microelement rotate around the z axis by the axial torque balance equations and the connection between the torsion and its angle.

$$\frac{\partial M_z}{\partial z} = -\rho_p J_p \frac{\partial \dot{\theta}_z}{\partial t} - \frac{M_y}{R} \tag{13}$$

$$\frac{\partial \dot{\theta}_z}{\partial z} = -\frac{1}{GJ_p}\frac{\partial M_z}{\partial t} - \frac{\dot{\theta}_y}{R} \tag{14}$$

where $V$ is the fluid velocity; $P$ is the pressure; $\rho_f$ is the fluid density; $R$ is the inner radius of pipeline; $\tau_0$ is the fluid wall bounded shear stress; $K$ is the fluid elastic modulus; $K^*$ is the modified fluid elastic modulus; $v$ is the Poisson's ratio; $u$ is the pipeline displacement; $r$ is the average radius of pipeline; $E$ is the pipeline elastic modulus; $e$ is the thickness of pipeline; $f$ is force acted on pipeline cross section; $A$ is the section area of pipeline; $k$ is the shear coefficient; $\theta$ is the pipeline angle; $M$ is the pipeline torque; $I$ is the pipeline lateral inertia; $J$ is the pipeline rotary inertia; $G$ is the shear modulus; $\Psi$ is the bending angle of pipeline; $ff$ is the elastic correction factor. The subscript $f$ represents the fluid; $p$ indicates the pipeline; $x$, $y$, and $z$ indicate the direction.

The modified coefficient of elasticity is expressed as

$$ff = \frac{1.65r^2}{eR} \tag{15}$$

*2.2. Friction Term of FSI 14-Equation Model*

The friction coupling of hydraulic pipe shows the influence of the shear stress generated by fluid flow on the dynamic behavior of structure coupling. The continuity equation and the Navier-Stokes (N-S) equation together derive the classic water hammer equations, which are the base of Equation (23) and Equation (15). The friction term in the equation can be described as Equation (27)

$$f(t) = \frac{2\tau_0(t)}{\rho_f R} \tag{16}$$

Among them, $\tau_0$ is the shear frictional force of the fluid to the inner wall of pipeline, it can be observed that the friction coupling of the pipeline is only related to it.

*2.3. Friction Coupling Model*

According to fluid flow conditions, friction models can be classified into the steady state friction model and the unsteady state friction model. The steady state friction model is commonly used to one-dimensional and steady flow state, on the contrary, the unsteady state friction model is adopted when the fluid is under turbulent flow state. The unsteady state friction models can be divided into two types: the quasi-steady state friction model based on the quasi-steady assumption and the unconstant friction model.

The unconstant friction model includes one-dimensional and two-dimensional unconstant friction model. Depending on the mechanism, one-dimensional unconstant friction model mainly includes

three types: the empirical equation model represented by Brunone model based on instantaneous acceleration [23], the weighted function model represented by the Zielke model when considering the influence of historical speed and historical acceleration on the flow state [15] and the irreversible thermodynamic model represented by Axworthy model [27].

### 2.3.1. Steady State Friction Model

The average velocity, pressure, density, and other parameters of the fluid on each section of the pipeline do not change or change little with time when the fluid is flowing at steady state in pipeline. According to the Newton friction formula and Darcy formula, the shear friction force of the inner wall of pipeline can be represented as

$$\tau = \frac{4\rho_f v_f V}{R} \tag{17}$$

The steady state friction model is described as

$$f(t) = \frac{2\tau}{R\rho_f} = \frac{8v_f V}{R^2} \tag{18}$$

The steady state friction model is only applicable for one-dimensional, laminar, and steady flow, but not for turbulent flow.

### 2.3.2. Quasi-Steady State Friction Model

Based on the quasi-steady assumption: the wall shear stress in unsteady flow is approximately equal to that in steady flow, the Quasi-steady state friction model is built as follows

$$\tau_{ws} = \frac{f\rho_f V|V|}{8} \tag{19}$$

where $f$ is the friction coefficient, which is related to the fluid property, flow state and roughness of the inner wall of pipeline. In the laminar flow, it can be expressed as

$$f = \frac{64}{Re}, \ Re = \frac{VD}{v_f} \tag{20}$$

The quasi-steady state friction model in laminar flow is the steady state friction model.

In the unsteady flow, the velocity profile will be changed with time going on, which would lead to the friction coupling mechanism become more complex and the calculated results of quasi-steady state friction model exist more errors. Therefore, the unconstant friction model with high accuracy has been developed.

### 2.3.3. The Unconstant Friction Model

The unconstant friction model is comprised of quasi-steady friction term and additional friction term. In view of the restrictions to the application of the two-dimensional friction model and the irreversible thermodynamic model in one-dimensional friction model, the two types are no longer discussed in this paper.

(1) The empirical formulae model based on instantaneous acceleration

Daily considered that the friction coupling was related to unsteady velocity and instantaneous acceleration and obtained an initial unsteady friction model. Brunone thought that the friction coupling was related to mean velocity of the fluid $V$, instantaneous acceleration $\partial V/\partial t$ and convective acceleration $\partial V/\partial z$. Based on the Daily friction model, the classical Brunone model was presented and widely applied.

The shear friction force in the model is as follows

$$\tau_w = \tau_{ws} + \tau_{wu} \tag{21}$$

$$\tau_{ws} = \frac{\rho_f f V |V|}{8} \tag{22}$$

$$\tau_{wu} = \frac{\rho_f D}{4} \cdot k_3 \left( \frac{\partial V}{\partial t} - \alpha \frac{\partial V}{\partial z} \right) \tag{23}$$

where $\alpha$ is wave velocity of water hammer; $k_3$ is the Brunone friction coefficient. In numerical simulation, $k_3$ is an usually empirical value which is determined by trial or experiment, so its application is limited.

In the Brunone model, the friction coupling of pipelines are combined with the fluid characteristics, and the influence of the convection term on friction coupling is taken into consideration. At the same time, the pressure wave velocity is included in the friction model, so the unsteady friction coupling is simulated more truly. Based on the global acceleration hypothesis, the semi-experience and semi-theory calculating formula of $k_3$ in the Brunone friction model was given by Vardy as follows [18]

$$k_3 = \frac{\sqrt{7.41 / Re^{k_4}}}{2} \tag{24}$$

$$k_4 = \log \frac{14.3}{Re^{0.05}} \tag{25}$$

It has been proven by many researchers that this formula can be applied to the smooth turbulent hydraulic region, and can be used directly at time of solving the friction coupling of the pipeline owing to accurate calculation results, which could provide a good analysis on friction coupling under high Reynolds number conditions.

(2) The weight function model considering the influence of historical flow velocity and acceleration on the current flow regime

Based on the N-S equation, Zielke presented the weight function friction model related to frequency. The weight function which could reflect the impact of historical flow velocity and historical acceleration on the current flow state is added to the model. The shear friction force in this model is as follows

$$\tau_w = \tau_{ws} + \tau_{wu} \tag{26}$$

$$\tau_{ws} = \frac{4\rho_f v_f}{R} V(t) \tag{27}$$

$$\tau_{wu} = \frac{2\rho_f v_f}{R} \int_0^t \frac{\partial V}{\partial t}(t_1) W(t - t_1) \mathrm{d}t_1 \tag{28}$$

where $\tau_w$ is the wall shear friction force in the unsteady flow; $\tau_{wu}$ is the wall additional shear force term in the unsteady flow; $W$ is the weight function of the dimensionless time $t$, which reflects the influence of historical flow velocity and historical acceleration on the current shear friction force.

In the Zielke model, the current velocity information is achieved by doing the integral and weighting of historical information of the local velocity. The weight function is related to the calculation frequency, and the calculation results have high accuracy in the laminar flow. So, it is generally only used for laminar flow or turbulent flow with low Reynolds number and acknowledged as an accurate model in the frequency domain.

## 2.4. Modification of FSI Friction Term

The Zielke friction model is based on the Newton friction formula and various constant flow assumptions are used. The model is related to the local fluid acceleration and the change of the historical velocity. It is accurate, reliable, and widely used in laminar flow, however, the error is greater when

turbulent flow. The Brunone empirical model connected friction coupling with instantaneous local acceleration and convective acceleration. Vardy has established a semi-experience and semi-theory formula for empirical value $k_3$. In turbulent flow, especially in high Reynolds number, the accuracy of the Brunone model is high and the simulation results for unsteady flow are great. On the basis of analysis and research, the steady-state model, the Zielke model and the Bruneone friction model are combined to form a new shear friction force model applicable for a wide range of Reynolds number expressed as

$$\tau_w = \tau_{ws} + k_z \tau_{wuz} + k_b \tau_{wub} \tag{29}$$

where $\tau_{ws}$ is the wall shear friction in steady flow, $\tau_{wuz}$ is the Zielke additional shear friction term in unsteady flow, $\tau_{wub}$ is the Brunone additional shear friction term in unsteady flow, $k_z$ is the Zielke additional shear friction coefficient and $k_b$ is the Brunone additional shear friction coefficient.

Combining Equation (16) and Equation (29), the modified FSI friction model is given as

$$f(t) = \frac{2}{R\rho_f}\left[\frac{4\rho_f \nu_f V}{R} + k_z \frac{2\rho_f \nu_f}{R}\int_0^t \frac{\partial V}{\partial t}W\mathrm{d}t + k_b \frac{\rho_f D}{8}\left(\frac{\partial V}{\partial t} - \alpha\frac{\partial V}{\partial z}\right)\right] \tag{30}$$

The flow state of the fluid must be determined to ensure the additional shear friction coefficient in the friction model before selection of the friction item. In laminar flow, $k_z = 1$ and $k_b = 0$, instead, $k_z = 0$ and $k_b = k_3$.

## 3. The Frequency-Domain Transfer Matrix Method for Hydraulic Pipeline

The transfer matrix method has the advantages of high accuracy, short computation time and easy programming, so it is primarily used to solve the FSI 14-equation model in this paper.

The derivation process of the frequency-domain transfer matrix is attached as Appendix A.

## 4. Numerical Simulation and Discussions

To get the vibration characteristics of fluid-structure interaction of pipelines, and verify the correctness of the friction model, a left-wing hydraulic pipeline of the C919 airplane is taken as the research object. Because the friction coupling between fluid and pipeline mainly affects the axial vibration response of pipelines, the axial vibration characteristics of pipelines are mainly analyzed.

### 4.1. Simulation Analytical Model

The aircraft hydraulic pipeline is an aluminum alloy pipeline and is divided into five sections according to the constraints and solving methods, which is shown in Figure 2. The pipeline is modeled and solved in MATLAB. The precise parameters of the pipeline and fluid are listed in Table 1.

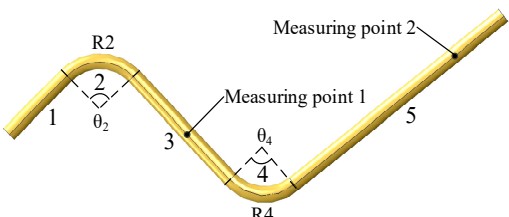

**Figure 2.** Aircraft hydraulic pipeline model.

Two measuring point are selected respectively in the middle section and the end section of the pipeline, the modal and the axial vibration velocity characteristics of the pipeline are carried out.

**Table 1.** The precise parameters of the pipeline and fluid.

| Parameter Name | Numerical Value | Parameter Name | Numerical Value |
|---|---|---|---|
| $L_1$ | 49.613 mm | pipeline density/$\rho_p$ | 7800 kg/m$^3$ |
| $L_3$ | 85.216 mm | Young's modulus/$E$ | 213 GPa |
| $L_5$ | 162.253 mm | shear modulus/$G$ | 81.9 GPa |
| $R_2$ | 28.575 mm | Poisson ratio/$v$ | 0.33 |
| $R_4$ | 28.575 mm | fluid density/$\rho_f$ | 872 kg/m$^3$ |
| thickness of pipeline/$e$ | 0.889 mm | Bulk modulus/$K$ | 1.95 GPa |
| bending angle $\theta_2$ | 1.595 rad | Kinematic viscosity/$v$ | 19.7 mm$^2$/s |
| bending angle $\theta_4$ | 1.480 rad | Material | 6061-T6 |
| inner radius of pipeline/$r$ | 3.874 mm | hydraulic oil | 10 # |

*4.2. Modal Simulation*

In order to study the fluid structure interaction between the pipeline and the fluid, and to verify the correctness of the friction model, the modal analysis is performed first. It is assumed that the pipeline is full of fluid, and the pipeline ends are sealed by screw plugs in the process of modal analysis. The two pipeline ends are free, and screw plugs are loaded at both ends as additional mass.

The mechanical shock excitation $F_r[1(t) - 1(t - T)]$ is applied to the initial end of the pipeline. The average impact force is $F_r = 200$ N, and the impact duration is $T = 2$ ms. After the Laplace transformation, the corresponding frequency domain excitation is $(F_r/s)(1 - e^{-sT})$, the excitation form of the pipeline can be expressed as

$$\begin{cases} \mathbf{Q}_0(s) = \begin{bmatrix} 0 & -(F/s)(1-e^{-sT}) & 0 & 0 & 0 & 0 & 0 \end{bmatrix}^{\mathrm{T}} \\ \mathbf{Q}_L(s) = \begin{bmatrix} 0 & 0 & 0 & 0 & 0 & 0 & 0 \end{bmatrix}^{\mathrm{T}} \end{cases} \tag{31}$$

Based on the above mechanical shock excitation and frequency-domain transfer matrix method, the axial vibration velocity response of the two measuring points on the pipeline are shown in Figure 3.

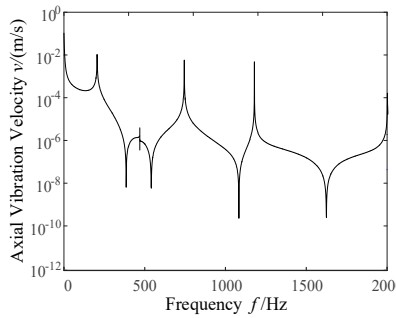

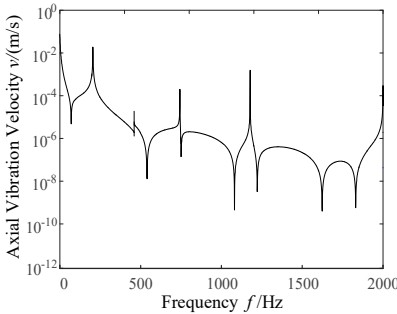

(**a**) Measuring point 1

(**b**) Measuring point 2

**Figure 3.** Axial velocity response curve of pipeline.

The first four simulation modes of the two measuring points are described in Table 2.

**Table 2.** The first four simulation modes of the two measuring points.

| Modal | 1st-Formant | 2nd-Formant | 3rd-Formant | 4th-Formant |
|---|---|---|---|---|
| Measuring point 1 | 209 Hz | 475 Hz | 751 Hz | 1182 Hz |
| Measuring point 2 | 207 Hz | 472 Hz | 748 Hz | 1183 Hz |

It can be observed from Figure 3 and Table 2 that there will be four formants from 0 Hz to 2000 Hz, and the corresponding axial vibration velocity value of the second formant is lower than the other formants obviously. Thus, it is reasonable to infer that the major vibration mode of the pipeline is not the axial vibration under the second-order natural frequency.

### 4.3. Axial Vibration Velocity Simulation

In order to simulate the high-pressure condition of the pipeline, the pressure of the aluminum alloy pipeline is setting and keeping at 12 MPa. The critical speed $V_c$ (5.90 m/s) of internal flow change is obtained according to the critical Reynolds number (2320) of the hydraulic pipeline. Hence, the flow velocity, flow rate and corresponding Reynolds number are listed in Table 3.

**Table 3.** The flow velocity, flow rate and corresponding Reynolds number.

| Parameter | Value | | | |
|---|---|---|---|---|
| Flow velocity/m/s | 2 | 4 | 6 | 8 |
| Flow rate/L/min | 5.66 | 11.32 | 16.97 | 22.63 |
| Reynolds number | 786.60 | 1573.20 | 2359.80 | 3146.40 |

The periodic pulsating fluid at the exit of an axial piston pump is regarded as the excitation, and the instantaneous pulsating flow is

$$q_{sh} = \sum_{i=1}^{z_0} Av_i = AR\omega \tan\gamma \sum_{i=1}^{z_0} \sin\varphi_i \tag{32}$$

In the formula, $A$ is the cross-sectional area of the plunger; $\gamma$ is the rotating speed of the cylinder; $z_0$ is the plunger number in the oil pressure area at the same time; $\varphi_i$ is the angle of the $i$ plunger in the oil pressure area relative to the top dead center; $v_i$ is the axial velocity of the $i$ plunger in the oil pressure area.

Without considering the effect of piston pump speed on pipeline vibration, the motor speed is chosen as 1000 r/min, the frequency domain curves of the flow rate of piston pump are shown in Figure 4.

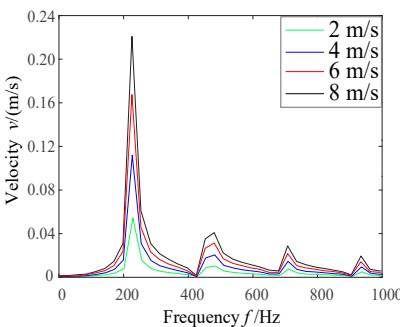

**Figure 4.** Frequency domain curve of flow pulsation of piston pump.

The first three order modals of the frequency domain flow pulsation curve are superimposed and the frequency domain flow excitation under each flow can be described as

$$\widetilde{q}_{shi} = \sum_{j=1}^{3} q_{ij}\omega_{ij}/\sin(s^2 + \omega_{ij}^2); i = 1, 2, 3, 4 \tag{33}$$

$q_{ij}$ is frequency domain fluid pulsating excitation coefficient as shown below

$$\begin{bmatrix} q_{11} & q_{12} & q_{13} \\ q_{21} & q_{22} & q_{23} \\ q_{31} & q_{32} & q_{33} \\ q_{41} & q_{42} & q_{43} \end{bmatrix} \begin{bmatrix} \omega_{11} & \omega_{21} & \omega_{31} \\ \omega_{12} & \omega_{22} & \omega_{32} \\ \omega_{13} & \omega_{23} & \omega_{33} \\ \omega_{14} & \omega_{24} & \omega_{34} \end{bmatrix}^{\mathrm{T}} = \begin{bmatrix} 0.05474 & 0.00992 & 0.00693 \\ 0.10933 & 0.01982 & 0.01383 \\ 0.16409 & 0.02974 & 0.02076 \\ 0.21842 & 0.03959 & 0.02763 \end{bmatrix} \begin{bmatrix} 1430.6 & 3040.1 & 4470.7 \\ 1430.6 & 3040.1 & 4470.7 \\ 1430.6 & 3040.1 & 4470.7 \\ 1430.6 & 3040.1 & 4470.7 \end{bmatrix}^{\mathrm{T}} \tag{34}$$

The fluid pulsation of the piston pump is regarded as the boundary condition, the fluid pulsation excitation of the pipeline is $\mathbf{V} = \mathbf{Q}/Af$. After the Laplace transformation, the corresponding excitation vector can be obtained as

$$\begin{cases} \mathbf{Q}_0(s) = \begin{bmatrix} V_e & 0 & 0 & 0 & 0 & 0 & 0 \end{bmatrix}^{\mathrm{T}} \\ \mathbf{Q}_L(s) = \begin{bmatrix} 0 & 0 & 0 & 0 & 0 & 0 & 0 \end{bmatrix}^{\mathrm{T}} \end{cases} \tag{35}$$

When the pressure is 12 MPa, the temperature is 20 °C, and the above fluid pulsation excitation is applied, the axial vibration velocity response of the pipeline is shown from Figures 5–8.

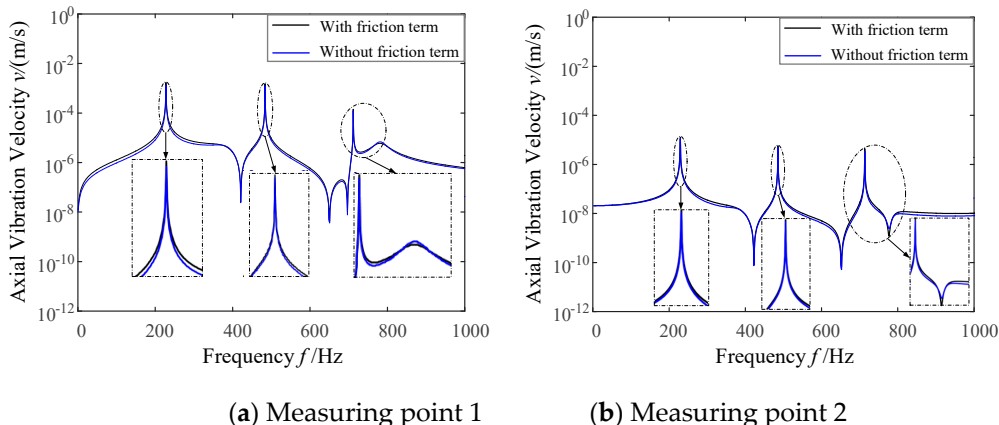

(**a**) Measuring point 1          (**b**) Measuring point 2

**Figure 5.** Axial velocity response when the flow velocity is 2 m/s.

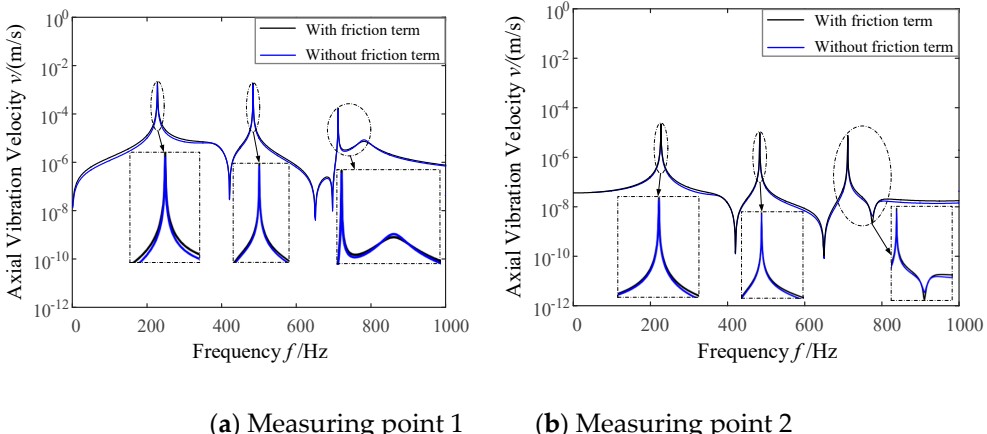

(**a**) Measuring point 1          (**b**) Measuring point 2

**Figure 6.** Axial velocity response when the flow velocity is 4 m/s.

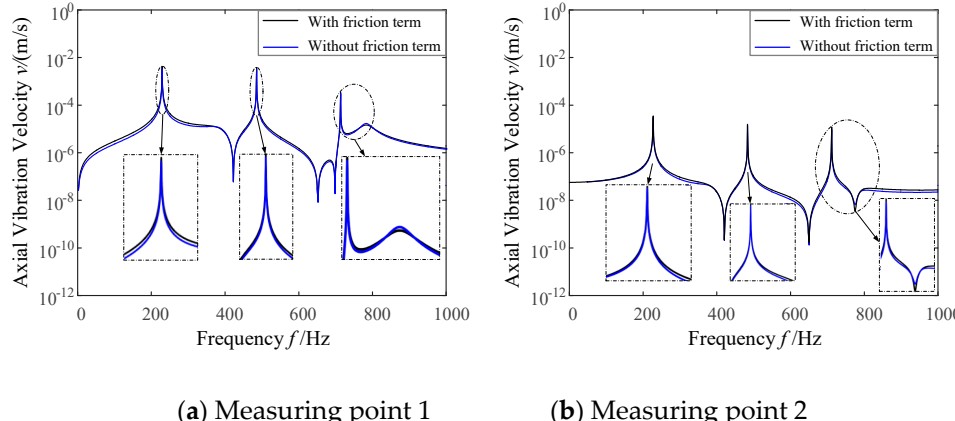

(**a**) Measuring point 1　　　　　(**b**) Measuring point 2

**Figure 7.** Axial velocity response when the flow velocity is 6 m/s.

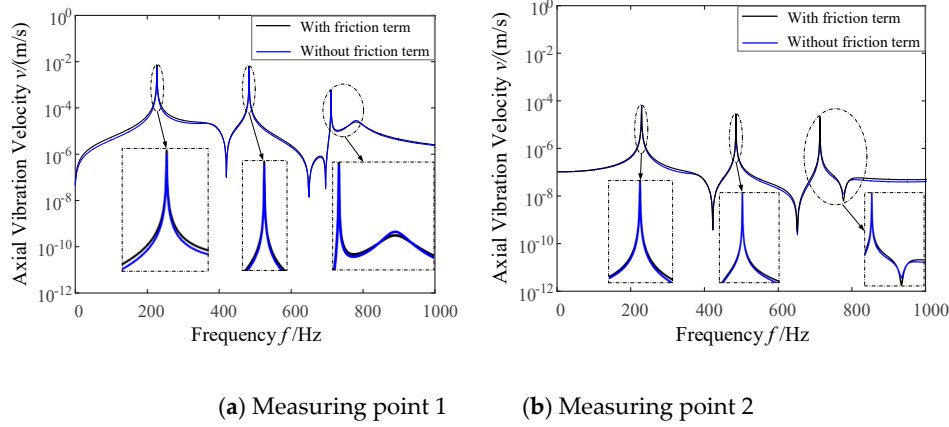

(**a**) Measuring point 1　　　(**b**) Measuring point 2

**Figure 8.** Axial velocity response when the flow velocity is 8 m/s.

Between the comparison from Figures 5–8, we can see that the axial vibration velocity response curves have the same trend with or without friction term. However, several points are worth noting.

(1) There will be corresponding peak value of resonance in the axial vibration velocity response of each position of the pipeline at each natural frequency of the pipeline. The friction term will not affect the resonance frequency of the pipeline under the fluid-structure interaction, while the response amplitude of each resonant frequency is impacted slightly, the greater the influence will be in the high frequency area.

(2) The amplitude of axial vibration velocity response will be raised with the increase of flow rates when the pressure is constant inside the pipeline, while the increase in the flow rate will not have an impact on the general trend of the frequency domain characteristics curve.

(3) The response amplitude of each resonant mode of measuring point 2 is higher than that of measuring point 1. Combined with Figure 2, we found that the measuring point 1 is located in the middle of the pipeline and the measuring point 2 is in the end of the pipeline, hence, it is reasonable to infer that the closer to the fixed end of the pipeline, the less affected by the fluid-structure interaction.

## 5. Experimental Validation and Discussions

In order to further study the FSI vibration characteristics of aircraft hydraulic pipeline and verify the correctness of the friction model, an experimental study is given. The experiment is performed on the test bench system of aircraft hydraulic pipeline vibration (Figure 9), the axial vibration response of the pipeline is measured by the Danish BK4525-B-001 acceleration sensor, its measuring range is about ±700 m/s$^2$, and 20 kHz, the NI measurement and control system is used for data acquisition and processing.

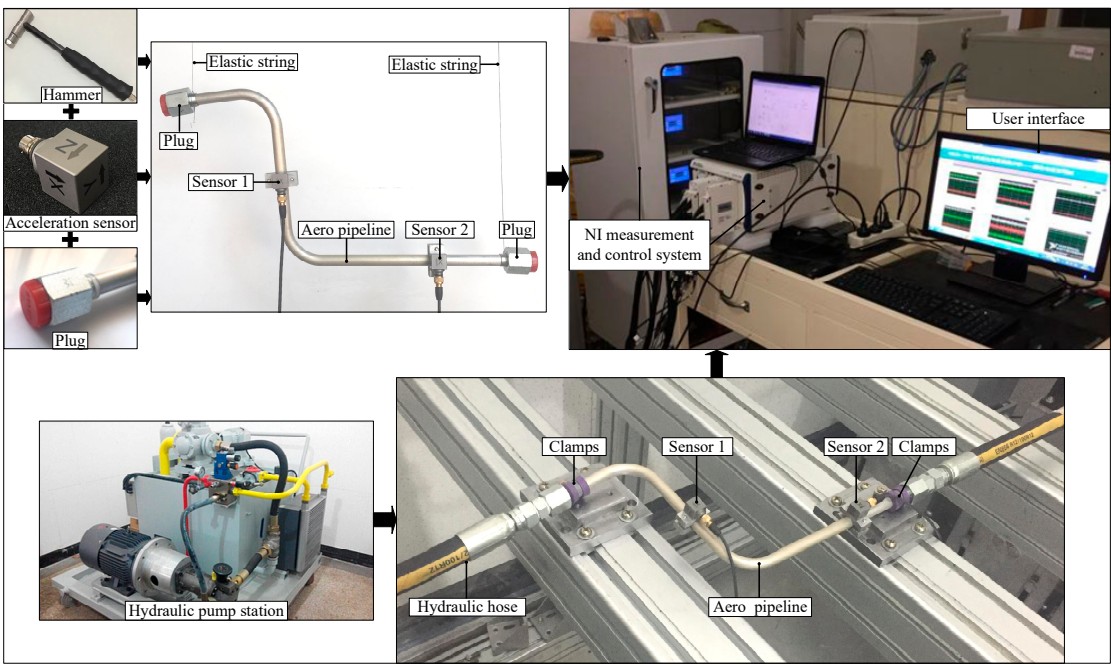

**Figure 9.** The overall experimental apparatus.

## 5.1. Modal Experiment

Based on the basic principle of modal analysis, the method of force-hammer excitation was adopted in the experiment. The pipeline is full of fluid and sealed by screw plugs, and it is suspended in the air with an elastic string to simulate the free state of the pipeline. The pipeline is hammered by the force hammer, causing the pipeline excited by pulse and the impact direction of the pulse is the radial direction of the pipeline. The vibration response of the pipeline will occur under the ideal unit impulse excitation. The amplitude of the harmonic component near the natural frequency is the largest, so the natural frequency can be obtained according to the frequency response.

The signal of the force hammer is shown in Figure 10, the excitation bandwidth satisfies the test require and can be considered as the ideal pulse excitation signal.

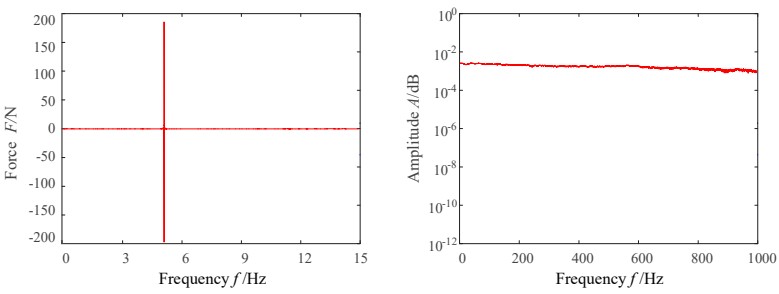

(**a**) Time domain signal      (**b**) Frequency domain signal

**Figure 10.** The signal curve of the force hammer.

The axial vibration velocity response of the two measuring points on the pipeline based on the above excitation signal is shown in Figure 11.

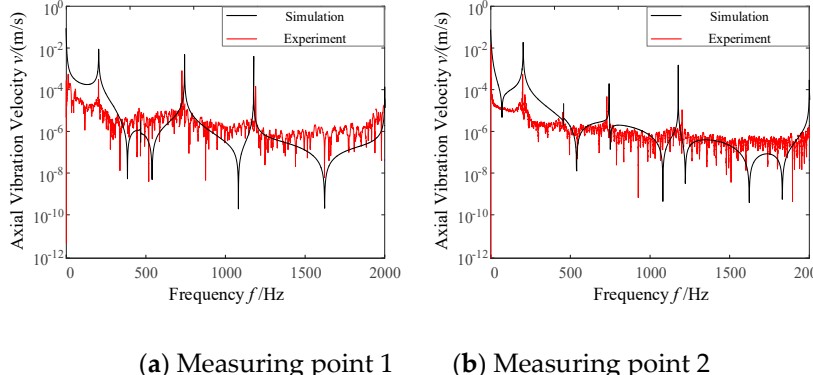

(**a**) Measuring point 1          (**b**) Measuring point 2

**Figure 11.** Axial velocity response curve of pipeline.

The first four experimental modes of the two measuring points and the error compared with simulation are described in Table 4.

**Table 4.** The first four experimental modes of the two measuring points.

|  | 1st-Formant | 2nd-Formant | 3rd-Formant | 4th-Formant |
|---|---|---|---|---|
| Measuring point 1 | 206 Hz | 465 Hz | 737 Hz | 1190 Hz |
| Error compared with simulation | 1.4% | 2.1% | 1.9% | 0.68% |
| Measuring point 2 | 204 Hz | 463 Hz | 739 Hz | 1202 Hz |
| Error compared with simulation | 1.4% | 1.9% | 1.5% | 1.6% |

The comparison between Tables 2 and 4 shows that the maximum error value between simulation and experiment is less than 2.1%, the good consistency of simulation and experiment could indicate that the modals and the solution process were correct and valid.

### 5.2. Axial Vibration Velocity Experiment

An experimental study on the FSI axial vibration velocity response of pipeline in different flow states was carried out. The pipeline is attached to the test bench system through hydraulic hoses, the pipeline ends are fixed on the bench by P-type clamp, the axial vibration response of different positions is measured by the acceleration sensor, as shown in Figure 9.

The speed of piston pump is set to 1000 r/min and the system pressure is 12 MPa. By changing the displacement of the piston pump, the flow velocity is 2 m/s, 4 m/s, 6 m/s and 8 m/s respectively. The axial vibration response of the position 1 and position 2 is measured. The axial vibration velocity response of the pipeline is shown from Figures 12–15.

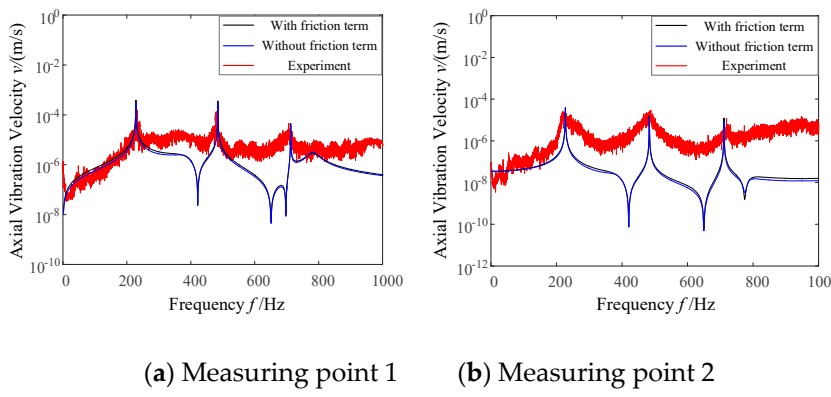

(**a**) Measuring point 1          (**b**) Measuring point 2

**Figure 12.** Axial velocity response curve when the flow rate is 2 m/s.

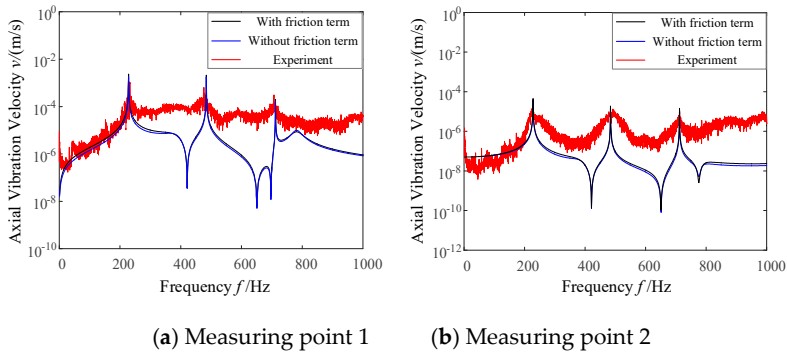

(**a**) Measuring point 1  (**b**) Measuring point 2

**Figure 13.** Axial velocity response curve when the flow rate is 4 m/s.

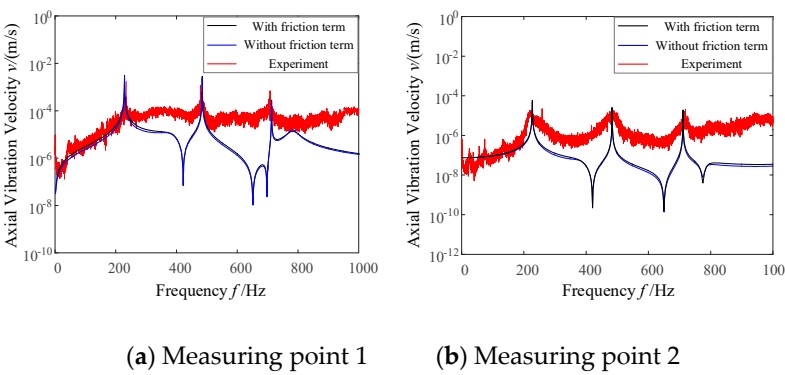

(**a**) Measuring point 1  (**b**) Measuring point 2

**Figure 14.** Axial velocity response curve when the flow rate is 6 m/s.

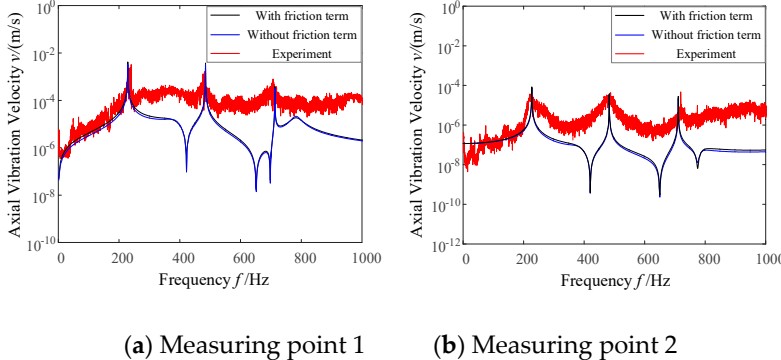

(**a**) Measuring point 1  (**b**) Measuring point 2

**Figure 15.** Axial velocity response curve when the flow rate is 8 m/s.

The axial velocity response of the pipe is as shown in Tables 5 and 6.

**Table 5.** The axial velocity response of the measuring point 1.

|  | Flow Rate |  | 1st-Formant | 2nd-Formant | 3rd-Formant |
|---|---|---|---|---|---|
| Simulation results | 2 m/s | 4 m/s | 230 | 486 | 712 |
|  | 6 m/s | 8 m/s |  |  |  |
| Experimental results | 2 m/s |  | 232.05 | 480.21 | 707.33 |
| Error |  |  | 0.81% | 1.23% | 0.70% |
| Experimental results | 4 m/s |  | 234.00 | 477.71 | 709.55 |
| Error |  |  | 1.65% | 1.74% | 0.38% |
| Experimental results | 6 m/s |  | 235.59 | 481.30 | 709.20 |
| Error |  |  | 2.34% | 1.01% | 0.43% |
| Experimental results | 8 m/s |  | 235.71 | 480.56 | 710.14 |
| Error |  |  | 2.39% | 1.16% | 0.30% |

**Table 6.** The axial velocity response of the measuring point 2.

| | Flow Rate | | 1st-Formant | 2nd-Formant | 3rd-Formant |
|---|---|---|---|---|---|
| Simulation results | 2 m/s 6 m/s | 4 m/s 8 m/s | 228 | 484 | 712 |
| Experimental results | 2 m/s | | 220.89 | 487.47 | 720.73 |
| Error | | | 3.32% | 0.66% | 1.22% |
| Experimental results | 4 m/s | | 226.42 | 496.20 | 712.44 |
| Error | | | 0.89% | 2.47% | 0.06% |
| Experimental results | 6 m/s | | 222.06 | 487.47 | 716.99 |
| Error | | | 2.80% | 0.66% | 0.70% |
| Experimental results | 8 m/s | | 223.28 | 485.90 | 716.94 |
| Error | | | 2.27% | 0.34% | 0.69% |

Tables 5 and 6 show that there is little disparity between simulation and experiment, and the maximum error is less than 3.32%, it may be caused by several reasons.

(1) There is residual stress in the pipeline because of the manufacturing error and installation error.

(2) Despite the impact direction of exterior excitation is the radial direction of the pipeline, there will be other directions excitation. Hence, other directions vibration might be measured by acceleration sensor, this is why the experimental curves are not as smooth as the simulation curves.

(3) Global error may cause by material parameters error, calculation error, etc.

Though there are little tolerance between simulation and experiment, the results are consistent and acceptable. There are also several points worthy to be discussed.

(1) The obvious valley value of axial vibration velocity is not obtained in the experiment. This is due to that the lower threshold of axial vibration velocity, which is the valley value of the axial vibration velocity of the FSI vibration of the pipeline, is set to $10^{-12}$ m/s when using a computer for simulation. However, the three-axis acceleration sensor only used to measure the peak value, but not to valley value in the experiment.

(2) The amplitude of axial vibration velocity response obtained by experiment is smaller than that obtained by numerical simulation at the same resonance frequency. This is because only the damping effect of fluid friction term is considered in numerical simulation, but there will be multiple damping and external signal interference of the system in the process of experiment.

(3) The general trend of simulation curves come closer to the experimental curves in consideration of friction term, this indicates that considering friction term can improve the prediction accuracy of FSI dynamic characteristics of the hydraulic pipeline.

## 6. Conclusions

In this paper, a fluid shear stress model of the pipe is presented to modify the friction model of the FSI 14-equation for high-speed and high-pressure hydraulic pipeline, then, the modal and axial vibration characteristics of an aircraft hydraulic pipeline is analyzed. Based on the research, the following conclusions can be drawn.

The fluid shear stress model is proposed based on the Brunone empirical model and the Zielke weighting function after analyzing and comparing the applicability of the various friction coupling models, which is suitable for the friction coupling analysis of hydraulic pipeline for a wide range of Reynolds number. The frequency-domain transfer matrix method is used to solve the FSI 14-equation model and to obtain the numerical simulation results of the modal analysis and axial vibration velocity of the hydraulic pipeline. The research result verify the correctness of the friction model developed, and indicate that the frequency-domain transfer matrix method could accurately analyze the formant frequency and axial vibration characteristics of fluid-structure interaction vibration of the hydraulic pipeline.

Of the simulation and experimental results in summary, when the pressure inside the pipeline is constant, the axial vibration velocity response amplitude increases with the increase of the flow

velocity, while the general trend of the frequency domain characteristic curve keeps unchanged. This indicates that the friction term will not affect the resonance frequency of the pipeline, but the response amplitude of each resonant frequency is impacted slightly. At each natural frequency of the pipeline, the corresponding resonance peak will appear in the axial vibration speed response of each position of the pipeline. Comparing the axial vibration velocity of different measuring points in the pipeline shows that the closer to the fixed end of the pipeline, the smaller influence of fluid-structure interaction on the pipeline. The general trend of simulation curves come closer to the experimental curves when considering friction term, this denotes that considering friction term can enhance the analytic accuracy of FSI dynamic characteristics of the hydraulic pipeline. Hence, it is of great practical meaning to consider the friction coupling effect when analyzing the FSI vibration characteristics of high-speed high-pressure hydraulic pipelines.

However, the calculation error in the process of simulation, the manufacturing error and installation error of the experimental pipeline, the multiple damping and external signal interference of the system cause small error to the final detection results. Further research will be carried out in the mixing region and the hydraulic rough region with higher Reynolds number in the future to improve the solution accuracy of the FSI 14-equation model.

**Author Contributions:** Conceptualization, L.Q. and M.G.; methodology, H.G. and S.C.; software, M.G. and H.G.; validation, L.Q., M.G. and H.G.; formal analysis, C.G.; investigation, C.G.; resources, L.Q.; data curation, M.G. and H.G.; writing—original draft preparation, H.G.; writing—review and editing, H.G.; visualization, M.G.; supervision, C.G.; project administration, L.Q.; funding acquisition, L.Q. All authors have read and agreed to the published version of the manuscript.

**Funding:** This research was funded by the National Key Research and Development Program of China, grant number (2014CB046400) and the National Natural Science Foundation of China, grant number (51775477 and 51505410).

**Acknowledgments:** The authors gratefully acknowledge the support of the above fundings and the authors also thank China Scholarship Council for supporting a two-years research experience of the first author and the corresponding author at the RWTH Aachen University and Washington State University. Meng Guo acknowledges the support of School of Mechanical Engineering, Beijing Institute of Technology.

**Conflicts of Interest:** The authors declare no conflict of interest.

**Appendix A. The Derivation Process of the Frequency-Domain Transfer Matrix in Section 3**

Following the derivation given in Refs. [11,12], the FSI 14-equation model is rewritten in matrix form as

$$\mathbf{A}(\partial/\partial t)\mathbf{\Phi}(z,t) + \mathbf{B}(\partial/\partial z)\mathbf{\Phi}(z,t) + \mathbf{C}\mathbf{\Phi}(z,t) = \mathbf{r}(z,t) \tag{A1}$$

where, matrix **A**, **B** are the 14-order constant term matrix of time differential and spacial differential respectively, represents the gradient of each variable to time and space. **C** is the friction and viscous damping coefficients, vector $\mathbf{r}(z,t)$ is the external excitation of the pipeline, $\mathbf{\Phi}(z,t)$ is the variable vector of the axial section position of the pipeline, and its expression is as follows

$$\mathbf{\Phi}(z,t) = \left(V, P, \dot{u}_z, f_z, \dot{u}_y, f_y, \dot{\theta}_x, M_x, \dot{u}_x, f_x, \dot{\theta}_y, M_y, \dot{\theta}_z, M_z\right)^{\mathrm{T}} \tag{A2}$$

Equation (A3) can be obtained via Laplace transform on the Equation (A1)

$$s\mathbf{A}^*(s)\widetilde{\mathbf{\Phi}}(z,s) + \mathbf{B}(\partial/\partial z)\widetilde{\mathbf{\Phi}}(z,s) = \tilde{\mathbf{r}}(z,s) + \mathbf{A}\mathbf{\Phi}(z,0) \tag{A3}$$

The expression of $\mathbf{A}^*(s)$ is as follows

$$\mathbf{A}^*(s) = \mathbf{A} + \mathbf{C}/s \tag{A4}$$

The expression of $\widetilde{\mathbf{\Phi}}(z,s)$ is as follows

$$\widetilde{\mathbf{\Phi}}(z,s) = \mathbf{S}(s)\widetilde{\mathbf{\eta}}(z,s) \tag{A5}$$

Then, Equation (A3) can be simplified as

$$s\widetilde{\mathbf{\eta}}(z,s) + \mathbf{\Lambda}(s)(\partial/\partial z)\widetilde{\mathbf{\eta}}(z,s) = s\widetilde{\mathbf{\eta}}_r(z,s) \tag{A6}$$

where $\mathbf{\Lambda}(s)$ and $\widetilde{\mathbf{\eta}}_r(z,s)$ are as follows

$$\mathbf{\Lambda}(s) = \mathbf{S}^{-1}(s)\mathbf{A}^{*-1}(s)\mathbf{B}\mathbf{S}(s) \tag{A7}$$

$$\widetilde{\mathbf{\eta}}_r(z,s) = (1/s)\mathbf{S}^{-1}(s)\mathbf{A}^{*-1}(s)\{\mathbf{r}(z,s) + \mathbf{A}\mathbf{\Phi}(z,0)\} \tag{A8}$$

where $\mathbf{\Lambda}(s)$ is a diagonal matrix composed of the eigenvalues of $\mathbf{A}^{*-1}(s)\mathbf{B}$, that is

$$\det(\mathbf{B} - \lambda(s)\mathbf{A}^*(s)) = 0 \tag{A9}$$

$$\mathbf{\Lambda}(s) = \begin{pmatrix} \lambda_1(s) & & & \\ & \lambda_2(s) & & \\ & & \ddots & \\ & & & \lambda_{14}(s) \end{pmatrix} \tag{A10}$$

where $\mathbf{S}(s)$ is the eigenvector matrix of $\mathbf{A}^{*-1}(s)\mathbf{B}$

$$\mathbf{S}(s) = (\xi_1(s), \xi_2(s), \cdots, \xi_{14}(s)) \tag{A11}$$

The general solution form of Equation (A6) is

$$\widetilde{\mathbf{\eta}}(z,s) = \mathbf{E}(z,s)\widetilde{\mathbf{\eta}}_0(s) + \widetilde{\mathbf{\eta}}_r^*(z,s) \tag{A12}$$

In Equation (A12), $\widetilde{\mathbf{\eta}}_0(s)$ is a constant related to boundary conditions, it could be written as

$$\mathbf{E}(z,s) = \begin{pmatrix} e^{-sz/\lambda_1(s)} & & & \\ & e^{-sz/\lambda_2(s)} & & \\ & & \ddots & \\ & & & e^{-sz/\lambda_{14}(s)} \end{pmatrix} \tag{A13}$$

$$\widetilde{\eta}_{ri}^*(z,s) = \frac{se^{-sz/\lambda_i(s)}}{\lambda_i(s)} \int^z \widetilde{\eta}_{ri}(z^*,s)e^{-sz^*/\lambda_i(s)}\mathrm{d}z^*, i = 1,2,\cdots,14 \tag{A14}$$

When $\widetilde{\eta}_{ri}^*(z,s)$ is not related to z, then $\widetilde{\eta}_{ri}^*(z,s)$ is equal to $\widetilde{\eta}_{ri}(z,s)$.

For axially distributed pipeline systems, $\widetilde{\mathbf{r}}(z,s) = 0$, that is $\mathbf{\Phi}(z,0) = 0$. According to Equation (A8), Equation (A3) and Equation (A14), $\widetilde{\mathbf{\eta}}_r(z,s) = \widetilde{\mathbf{\eta}}_r^*(z,s) = 0$. Substituting the above vectors into Equation (A11) gives

$$\widetilde{\mathbf{\eta}}(z,s) = \mathbf{E}(z,s)\widetilde{\mathbf{\eta}}_0(s) \tag{A15}$$

Then combined with Equation (A5) yield

$$\widetilde{\mathbf{\Phi}}(z,s) = \mathbf{S}(s)\mathbf{E}(z,s)\widetilde{\mathbf{\eta}}_0(s) \tag{A16}$$

When $z = 0$, $\mathbf{E}(0,s) = \mathbf{I}$

$$\widetilde{\mathbf{\Phi}}(0,s) = \mathbf{S}(s)\widetilde{\mathbf{\eta}}_0(s) \tag{A17}$$

By substituting Equation (A16) to Equation (A15) gives

$$\widetilde{\boldsymbol{\Phi}}(z,s) = \mathbf{M}(z,s)\widetilde{\boldsymbol{\Phi}}(0,s) \tag{A18}$$

where $M(z,s)$ is the field transfer matrix of the pipeline, which can be expressed as

$$\mathbf{M}(z,s) = \mathbf{S}(s)\mathbf{E}(z,s)\mathbf{S}^{-1}(s) \tag{A19}$$

In order to obtain the initial end state variables of pipeline, boundary conditions must be set at both ends of the pipeline. Assuming the pipeline length is $L$, there are seven boundary conditions at each end of the pipeline ($z = 0$ and $z = L$) as follows

$$[\mathbf{D}_0(s)]_{7\times14}\left[\widetilde{\boldsymbol{\Phi}}(0,s)\right]_{14\times1} = [\mathbf{Q}_0(s)]_{7\times1} \tag{A20}$$

$$[\mathbf{D}_L(s)]_{7\times14}\left[\widetilde{\boldsymbol{\Phi}}(L,s)\right]_{14\times1} = [\mathbf{Q}_L(s)]_{7\times1} \tag{A21}$$

where $\mathbf{D}_0(s)$ and $\mathbf{D}_L(s)$ are the boundary matrices with dimensions of $7 \times 14$, $\mathbf{Q}_0(s)$ and $\mathbf{Q}_L(s)$ are the excitation matrices with dimensions of $7 \times 1$ at each end of the pipeline.

According to Equation (A18), the transfer relationship between the state vectors at both ends of the pipeline is

$$\widetilde{\boldsymbol{\Phi}}(L,s) = \mathbf{M}(L,s)\widetilde{\boldsymbol{\Phi}}(0,s) \tag{A22}$$

The initial end state vectors of the pipeline are as follows

$$\widetilde{\boldsymbol{\Phi}}(0,s) = \mathbf{D}*^{-1}(s)\mathbf{Q}(s) \tag{A23}$$

where

$$\mathbf{D}^*(s) = \begin{pmatrix} \mathbf{D}_0(s) \\ \mathbf{D}_L(s)\mathbf{M}(L,s) \end{pmatrix} \tag{A24}$$

$$\mathbf{Q}(s) = \begin{pmatrix} \mathbf{Q}_0(s) \\ \mathbf{Q}_L(s) \end{pmatrix} \tag{A25}$$

When solving the FSI 14-eqation model of the pipeline, the overall field transfer matrix needs to be established.

$$\mathbf{M}_{\text{global}}(L,s) = \mathbf{M}_N(L_N,s)\cdots\mathbf{M}_i(L_i,s)\cdots\mathbf{M}_1(L_1,s) \tag{A26}$$

where $L_i$ is the length of each pipeline section, $\mathbf{M}_i$ is the field transfer matrix of each pipeline section.

According to Equation (A23), Equation (A26) and Equation (A18), the state variables at any position of the pipeline can be obtained.

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
