# Peer review of "Axial Vibration Characteristics of Fluid-Structure Interaction of an Aircraft Hydraulic Pipe Based on Modified Friction Coupling Model"

_applsci, doi:10.3390/app10103548_

Round 1

Reviewer 1 Report

The authors include improved frictional coupling in fluid-structure interaction of a hydraulic pipe. Both simulation and laboratory results are presented and the results seem in agreement. 

Minor typoes are noted such as in some headings of Tables 5 and 6, a formant is misspelled as "ormant".

Reviewer 2 Report

I found this work very fascinating from numerical point of view considering the complete 14-equation model in order to model FSI. In my opinion the introduction is over stated and could be shorter, but it can be a good reference for who start to work in this field.

I accept the paper as its current form based on my review and evaluation.

Reviewer 3 Report

See pdf document

Author Response

This manuscript is a resubmission of an earlier submission. The following is a list of the peer review reports and author responses from that submission.

Round 1

Reviewer 1 Report

Dear authors,

I would like first of all to commend your effort on a very interesting topic, well done.

I believe the manuscript can be further improved by incorporating a few changes below:

Significant editing of English language is required. Many times the use of gerund and past participle is confused.

At line 210 I believe that the 'dimensionless time' should not be represented by 'τ' as this has been allocated already to the shear friction force.

Please use relevant references for the non widely known equations that you refer to the text.

Section 3 contains extensive derivations which you should consider to include on a separate appendix for example so as not to distract the reader.

Please use coloured versions of all the figures used, as it is sometimes very hard to distinguish what is what.

It is not clear how the developed model is validated. Are the results backed up by relevant literature for instance?

In general, I believe that the literature can be further enhanced, especially with regards to the 'discussion' section of the article, as per the previous point.

  With regards to the errors involved, it will be useful to include a short error assessment, describing the error sources and their quantitative contribution, ie error margins and confidence intervals. 

Reviewer 2 Report

The paper addresses the axial vibration of a hydraulic elastic pipe involving fluid-structure interaction. The inclusion of a quadratic velocity term take into consideration of turbulent flow.

To improve the paper, how does the nonlinearity term handled in the frequency-domain transfer matrix method should be highlighted. With the curved pipe involved, secondary flow is expected. Its influence will need to be accounted for or discussed. It is also better to describe the FSI 14-equation in terms of the physics it based and the assumption it invokes. The error involved in measurement and simulation need to be discussed more thoroughly.

On line 28, the word "quality" seems confusing. Does it refer to mass?